

# The impact of serum uric acid on the natural history of glomerular filtration rate: a retrospective study in the general population

Ying Xu[1], Xiang Liu[1], Xiaohe Sun[2] and Yibing Wang[3]

[1] Department of Nephrology, Shandong Provincial Hospital affiliated to Shandong University, Jinan, Shandong, China
[2] School of Medicine, Shandong University, Jinan, Shandong, China
[3] Department of Burns and Plastic Surgery, Shandong Provincial Hospital affiliated to Shandong University, Jinan, Shandong, China

## ABSTRACT

Serum uric acid (SUA) level has been proposed to have important connections with chronic kidney disease (CKD), while the impact of SUA level on the natural history of glomerular filtration rate (GFR) decline remains unknown. The present study aims to study the association of the SUA level with the GFR decline in a general population. Two thousand, seven hundred and eighty-nine subjects who visited the Health Checkup Clinic both at 2008 and 2013 were identified. A significant inverse correlation was observed between change in SUA from 2008–2013 (ΔSUA) and change in eGFR (ΔeGFR) during the same period. Multivariate regression analysis of ΔeGFR indicated that the increase in SUA over time were a negative predictor of the change in eGFR. Our result indicates that the decline of eGFR over years is larger in subjects with an increased SUA level, which helps to underline the importance of SUA level management in the context of kidney function preservation.

## INTRODUCTION

Uric acid (UA) is a poorly soluble end product of purine nucleotides degradation, and hyperuricemia is defined as serum uric acid (SUA) level >420 umol/L (7 mg/dl) in males and >360 umol/L (>6 mg/dl) in females (*Chuang et al., 2012*). Recent researches have suggested SUA to be an independent risk factor for cardiovascular disease (*Chuang et al., 2012*) and metabolic syndrome (MetS) (*Liu et al., 2014*; *Nejatinamini et al., 2015*). Hyperuricemia was also suggested to be associated with the onset of type 2 diabetes mellitus (DM) (*Miyake et al., 2014*) and stroke (*Qin et al., 2014*).

As long as the kidney is concerned, the pathophysiological role of SUA in the chronic kidney disease (CKD) has attracted a lot of nephrologists and there are many studies published. It has been established that in patients with gout and acute UA nephropathy, the treatment of hyperuricemia is of clinical benefits. For patients without gout and

Corresponding author
Yibing Wang, wyb0616@163.com

acute nephropathy, the first aspect is the relationship between hyperuricemia and CKD progression. Whether SUA is an independent risk factor or only a marker of the progression of CKD is still a matter of debate. In the past, hyperuricemia was thought to be simply a reflection of kidney damage because it was assumed that a decrease in kidney blood flow causes the decrease in UA clearance. However, high uric acid levels was reported to be associated with increased rates of glomerular filtration rate (GFR) decline in multiple cross sectional studies in CKD patients (*Ben-Dov & Kark, 2011*; *Satirapoj et al., 2010*). On the other hand, an association of SUA with CKD was not demonstrated in other studies. The elevation of SUA did not contribute to the progression of CKD in the Mild to Moderate Kidney Disease (MMKD) study, which only recruited non-diabetic patients (*Sturm et al., 2008*). The second aspect is whether hyperuricemia is a predictor of incidence of CKD. There was no significant association between SUA and the incidence of CKD during the 5-year follow-up in the Cardiovascular Health Study, which recruited 5,808 subjects (*Chonchol et al., 2007*). However, in type 2 DM, hyperuricemia has been reported to be correlated with the onset of proteinuria (*Tseng, 2005*) and renal dysfunction (*Bo et al., 2001*; *De Cosmo et al., 2015*). A recent meta-analysis that included 13 studies also reported that a relationship between increasing SUA level and new onset of CKD (defined by eGFR <60 mL/min/1.73 m$^2$) (*Li et al., 2014*).

While the conclusion is still uncertain, most studies were done study the relationship of hyperuricemia and CKD incidence/progression, it also remains unknown whether the SUA level is associated with the natural decline of GFR with aging. It is important to study the strategies that could prevent the incidence of CKD and preserve the GFR in general population. The present study aims to study the association of the SUA level with the GFR decline in a general population who visited the Health Checkup Clinic regularly, and our results suggested a role of SUA level in the GFR decline.

## METHODS

The protocol of this study was approved by the Clinical Investigation Ethics Committee of Shandong Provincial Hospital Affiliated Shandong Provincial Hospital (Reference No. 2015-040) and was conducted in strict adherence with the principles of the Declaration of Helsinki.

### Study population

This is a retrospective study. Computer-based data of Health Checkup Clinic, Shandong Provincial Hospital Affiliated to Shandong University (Jinan, China) were available from Jan 1st, 2008–Dec 31st, 2013. Subjects who visited the Heath Checkup at 2008 and 2013 were identified using ID numbers, birthdates and other identifiers. All subjects are >18 years old. This study excludes outpatient or clinical patients. Patients with acute kidney injury, amputation, heart failure, severe liver disease, infection disease, malignant disease and pregnancy were excluded. To confirm the accuracy, all the matches were visually inspected. Thus, we identified 2,789 adults, of which 205 subjects were excluded for incomplete records. This left 2,584 subjects (1,791 males and 793 females) for analysis. All subjects gave the oral consent that the Health Checkup Clinic of Shandong

Provincial Hospital could use the anonymous data for research purpose only when they visited at the first time.

In 2008 and 2013, all subjects completed a detailed questionnaire administered by trained interviewers on life style factors such as smoking, consumption of alcohol and physical activity. Weight and height were measured with light indoor clothes and without shoes. An automated sphygmomanometer was used to measure systolic and diastolic blood pressures (SBP/DBP) with subjects in the seated position in triplicate. A blood sample was taken after an overnight fast. Biomedical parameters including serum uric acid, serum creatinine (Scr), fasting plasma glucose (FPG) and lipids were measured using a biochemical auto-analyzer in the Clinical Laboratory of the Shandong Provincial Hospital affiliated to Shandong University using standardized procedures.

### Definition

Body mass index was calculated as weight (kg) divided by the square of the height (m). Estimated Glomerular Filtration Rate (eGFR) was calculated using Modification of Diet in Renal Disease (MDRD) formula for Chinese (*Kong et al., 2013*; *Ma et al., 2006*). According to a research published in 2013 that recruited 682 patients and 295 healthy adults, MDRD formula for Chinese is comparable to the CKD-EPI two-level race equation. They both performed better than the MDRD Study equation and the CKD-EPI four-level equation (*Kong et al., 2013*). The longitudinal eGFR changes ($\Delta$eGFR) were calculated as the eGFR at 2013 minus the eGFR at 2008 (eGFR13 − eGFR08). Hyperuricemia is defined as serum uric acid level >420 umol/L (7 mg/dl) in males and >360 umol/L (>6 mg/dl) females (*Chuang et al., 2012*).

### Data analysis

Statistical analysis was conducted by using SAS, version 9.4 (SAS Institute Inc). The graph was drafted using SPSS, version 22. Values of $p < 0.05$ were considered to be statistically significant. Kolmogorov-Smirnov test was used for normality distribution. Descriptive statistics are presented as means with standard deviations for normally distributed data, median with interquartile range (Q75, Q25) for non-normally distributed data, and frequencies with percentage for categorical data. Paired t-test were used for normally distributed continuous variables, while Wilcoxon Signed-Rank test were used for non-normally distributed data in comparison between data in year of 2008 and that in 2013. Univariate and multivariate linear regression analysis was employed to estimate the relationship between SUA level and eGFR change over 5 years.

## RESULTS

### Demographic characteristics

Table 1 shows the demographic characteristics in the cohort. The SUA level was significantly increased from year 2008–2013 (322.00 (382.00, 267.00) versus 352.00 (410.00, 295.00) umol/L, $p < 0.01$). While the decline of eGFR across the 5-year period was statistically significant (103.99 (113.31, 95.11)–101.19 (113.24, 101.19) mL/min/1.73 m$^2$, $p = 0.08$) but in a very small manner. There were no differences in the BMI, systolic

**Table 1 Clinical characteristics of study subjects.**

| | 2008 | 2013 | t or Z | p-value* |
|---|---|---|---|---|
| Age (years) | 47.84 ± 12.64 | – | – | – |
| Gender | | | | |
|    Male (%) | 1,791 (69.31) | – | – | – |
|    Female (%) | 793 (30.69) | – | – | – |
| Habitual drinker (%) | 582 (22.52) | – | – | – |
| Habitual smoker (%) | 572 (22.14) | – | – | – |
| SUA (umol/L) | 322.00 (382.00, 267.00) | 352.00 (410.00, 295.00) | −22.43 | <0.01 |
| eGFR (mL/min/1.73 m$^2$) | 103.99 (113.31, 95.11) | 101.19 (113.24, 101.19) | −5.49 | <0.01 |
| BMI (kg/m$^2$) | 24.86 ± 3.24 | 24.06 ± 2.46 | 1.68 | 0.09 |
| Systolic BP (mmHg) | 122.00 (135.00, 111.00) | 122.00 (136.00, 110.00) | −0.31 | 0.75 |
| Diastolic BP (mmHg) | 73.00 (81.00, 66.00) | 71.00 (79.00, 63.00) | −13.39 | <0.01 |
| FPG (mmol/L) | 4.99 (5.47, 4.67) | 5.32 (5.83, 4.93) | −23.95 | <0.01 |
| TC (mmol/L) | 5.14 (5.77, 4.49) | 5.19 (5.84, 4.61) | −5.70 | <0.01 |
| TG (mmol/L) | 1.25 (1.93, 0.84) | 1.30 (1.87, 0.92) | −2.23 | <0.05 |

**Notes:**

Data are presented as mean ± SD for normally distributed data, median (Q75, Q25) for non-normally distributed data or n (percentage) for categorical data.

* 2008 versus 2013.

SUA, serum uric acid; BMI, body mass index; FPG, fasting plasma glucose; TC, total cholesterol; TG, triglycerides.

blood pressure (BP). Fasting plasma glucose (FPG), serum total cholesterol (TC) and serum triglycerides (TG) were slightly increased in 2013 compared to 2008, and the diastolic BP was marginally decreased. The ratio of subjects with dipstick-positive proteinuria at 2008 and 2013 was 3.4% and 2.8% respectively.

## Relationship between longitudinal changes in SUV and eGFR

Univariate linear regression analysis did not indicate a significant correlation between baseline SUA at 2008 and the change in eGFR from 2008–2013 (ΔeGFR) ($r = 0.03$, $p = 0.09$). In contrast, a significant inverse correlation was observed between change in SUA from 2008–2013 (ΔSUV) and ΔeGFR during the same period as shown in Fig. 1 ($r = −0.37$, $p < 0.01$).

As listed in Table 2, gender, fasting plasma glucose (FPG), serum total cholesterol (TC), serum triglycerides (TG) and baseline eGFR were significant predictors of the ΔeGFR ($t' = 8.30$, $r = 0.15$, $r = 0.10$, $r = 0.09$, $r = −0.30$, respectively; $p < 0.01$). Drinking alcohol ($p = 0.95$) or smoking ($p = 0.98$), BMI ($p = 0.42$), systolic blood pressure (SBP) ($p = 0.87$), and diastolic blood pressure (DBP) ($p = 0.52$) were not associated with ΔeGFR.

## Determinants of longitudinal changes in eGFR (ΔeGFR) in people with increased SUA level and decreased SUA level over 5 years

Secondary analyses included a trichotomized variable defined by subject's SUA increase versus decline from year 2008–2013. For a changed SUA level <10 umol/L, either increased or decreased, it is not clinically meaningful since it could be the experiment error. Therefore, we divided the subjects into three groups: subjects with increased SUA

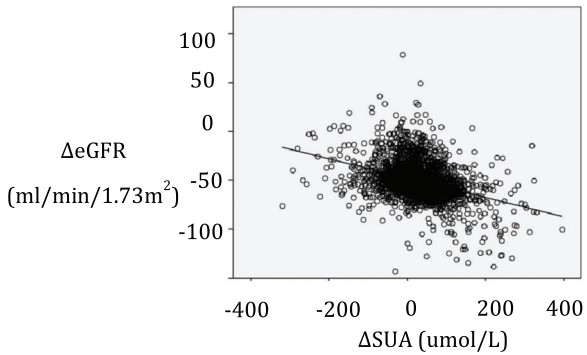

**Figure 1 Relationship between changes in SUA from 2008–2013 and changes in eGFR during the same period.** ΔSUA, change in the serum uric acid level from 2008–2013; ΔeGFR, change in estimated glomerular filtration rate from 2008–2013.

**Table 2 Univariate linear regression analysis for change in eGFR from 2008–2013 (ΔeGFR) and independent variables.**

|                 | $t'/r$   | $p$-value |
| --------------- | -------- | --------- |
| Gender          | 8.30     | <0.01     |
| Habitual drinker| −0.06    | 0.95      |
| Habitual smoker | −0.03    | 0.98      |
| SUA             | 0.03     | 0.09      |
| BMI             | 0.02     | 0.42      |
| Systolic BP     | −0.003   | 0.87      |
| Diastolic BP    | 0.01     | 0.52      |
| FPG             | 0.15     | <0.01     |
| TC              | 0.10     | <0.01     |
| TG              | 0.09     | <0.01     |
| eGFR at 2008    | −0.30    | <0.01     |
| ΔeGFR           | −0.37    | <0.01     |

(ΔSUA >10 umol/L), those with unchanged SUA (−10 umol/L < ΔSUA < 10 umol/L), and those with decreased SUA (ΔSUA < −10 umol/L). The demographic characteristics of threes groups were shown in Table 3. There were no differences in the distribution of gender, age, drink or smoke habit between three groups (all $p$-value >0.05). The baseline SUA is lower in the Increased SUA group. Together with SUA level, BMI, DBP, FPG and TG, were lower in the Increased SUA group compared with the other two groups. eGFR of the Increased SUA group was the highest. This indicates that the Increased SUA was at a "healthier" state at 2008. However, after 5 years, the eGFR of the Increased SUA group declined by 5.26 (3.71, −12.44) mL/min/1.73 m$^2$, and the eGFR of the Decreased SUA group incremented by 3.23 (15.25, −4.63) mL/min/1.73 m$^2$ with a $p$-value <0.01.

Based on the result of univariate linear regression analysis and the comparison between the three groups, we identified the independent variables that were put into multivariate linear regression model. They meet both criteria, first, $p$ of univariate linear regression

**Table 3 Clinical characteristics of three groups at baseline (2008).** Subjects with an increased SUA from 2008, with unchanged SUA, and with decreased SUA during the same period.

| | Increased SUA | Unchanged SUA | Decreased SUA | $F/\chi^2$ | $p$-value |
|---|---|---|---|---|---|
| Sex | | | | 1.41 | 0.49 |
|    Male | 1,154 | 214 | 423 | | |
|    Female | 500 | 108 | 185 | | |
| Age (years, 2008) | 47.50 ± 12.37 | 47.76 ± 12.95 | 48.80 ± 13.17 | 2.18 | 0.11 |
| Drink habit | | | | 2.27 | 0.32 |
|    Habitual | 379 | 62 | 141 | | |
|    Non-habitual | 1,275 | 260 | 467 | | |
| Smoke habit | | | | 1.46 | 0.48 |
|    Habitual | 378 | 69 | 125 | | |
|    Non-habitual | 1,276 | 253 | 483 | | |
| SUA (umol/L) | 310.00 (403.00, 253.00) | 315.00 (377.00, 267.00) | 377.50 (428.50, 318.00) | 284.40 | <0.01 |
| BMI (kg/m$^2$) | 24.65 ± 3.18 | 24.83 ± 3.37 | 25.28 ± 3.27 | 13.70 | <0.01 |
| SBP (mmHg) | 121.00 (135.00, 110.00) | 119.00 (132.00, 110.0) | 125.00 (138.00, 114.00) | 17.78 | <0.05 |
| DBP (mmHg) | 73.00 (81.00, 66.00) | 74.00 (80.50, 66.00) | 75.00 (84.00, 68.00) | 11.20 | <0.01 |
| FPG (mmol/L) | 4.97 (5.42, 4.66) | 5.00 (5.51, 4.66) | 5.07 (5.70, 4.73) | 17.40 | <0.01 |
| TC (mmol/L) | 5.10 (5.74, 4.48) | 5.17 (5.91, 4.56) | 5.21 (5.91, 4.56) | 5.57 | 0.05 |
| TG (mmol/L) | 1.22 (1.86, 0.82) | 1.18 (1.84, 0.80) | 1.41 (2.12, 0.92) | 23.50 | <0.01 |
| eGFR at 2008 (mL/min/1.73 m$^2$) | 105.16 (113.78, 96.23) | 103.69 (112.59, 95.97) | 101.87 (112.21, 91.36) | 29.55 | <0.01 |
| ΔeGFR (mL/min/1.73 m$^2$) | −5.26 (3.71, −12.44) | 0.00 (9.93, −6.64) | 3.23 (15.25, −4.63) | 191.39 | <0.01 |

**Note:**
Data are presented as mean ± SD for normally distributed data, median (Q75, Q25) for non-normally distributed data or n.

analysis <0.1, and second, there were differences among three groups. Together with the trichotomized variable defined by subject's SUA change, baseline SUA, FPG, TC, TG, and baseline eGFR were fitted in the multiple linear regression model. The collinearity diagnostics show that vif was close to 0, and condition index was >10, indicating that multicollinearity exists. However, there were no closely related variables detected. Using forward stepwise selection with inclusion criteria of $p < 0.05$ and exclusion criteria of $p > 0.10$, we get the multivariate regression analysis results as shown in Table 4. Multivariate regression analysis indicated that increased SUA together with baseline SUA level, baseline FPG, baseline TC, baseline TG, and baseline eGFR were negative predictor of ΔeGFR (ΔeGFR = eGFR at 2013 − eGFR at 2008) with $R^2$ of 0.17.

## DISCUSSION

The present study demonstrates that the change in SUA during a 5-year period is inversely associated with the decline of eGFR during the same period.

Chronic kidney disease (CKD) has become a global public health problem because of the high prevalence and the accompanying increase in the risk of end-stage renal disease (ESRD), cardiovascular disease and premature death (*Tonelli et al., 2006*). There are many cohorts that show the development and progression of CKD are linked with uric acid level. A Korean study showed that asymptomatic hyperuricemia men had a greater odd

**Table 4 Multiple linear regression analysis for change in eGFR from 2008–2013 (ΔeGFR) and independent variables.**

|  | $\beta$ | SE | $t$ | $p$-value |
|---|---|---|---|---|
| Intercept | 42.24 | 3.88 | 10.88 | <0.01 |
| ΔSUA |  |  |  |  |
|     Increased SUA | −11.52 | 0.90 | −12.79 | <0.01 |
|     Unchanged SUA | −5.32 | 1.24 | −4.28 | <0.01 |
|     Decreased SUA | 0 | 0 |  |  |
| SUA (umol/L) | −0.03 | <0.01 | −5.37 | <0.01 |
| FPG (mmol/L) | 0.14 | 0.03 | 4.06 | <0.01 |
| TC (mmol/L) | 0.83 | 0.38 | 2.15 | <0.05 |
| TG (mmol/L) | 0.99 | 0.32 | 3.11 | <0.01 |
| eGFR (mL/min/1.73 m$^2$) | −0.32 | 0.02 | −13.87 | <0.01 |

**Notes:**
$R^2 = 0.17$.
$\beta$, standardized regression coefficient. SE, standard error.

(odds ratio, 1.96) of developing CKD than normouricemic men (*Ryoo et al., 2013*). Epidemiologic data from a Japanese cohort showed that hyperuricemia was an independent risk factor for decrease in eGFR (*Iseki, Iseki & Kinjo, 2013*). New-onset of microalbuminuria was shown to be associated with hyperuricemia in a middle-age and elderly Taiwanese population (*Chang et al., 2013*). However, the highest uric acid level did not lead to an increased incidence of CKD according to an analysis of the MDRD Study population (*Madero et al., 2009*). In contrast, the effects of hyperuricemia on a natural history of GFR have been less examined. In the present study, we did find a difference in the decline of eGFR over five years between asymptomatic hyperuricemia subjects and normal SUA level subjects (data not shown). While a change in SUA is associated with the ΔeGFR, though an association could not indicate cause-effect relationship. Our results are in consistent with a recent study, which included two cohorts, urban residents and rural town residents respectively. Their results also indicate that the elevation of SUA accelerates the yearly decline in eGFR (*Akasaka et al., 2014*).

Interventions targeting at hyperuricemia may decrease disease progression in patients with CKD. In a trial of 96 patients with CKD stages 3–4 with mean eGFR of 44.62 ± 14.38 mL/min/1.73 m$^2$, allopurinol or control was randomly assigned. Those patients with lower serum uric acid levels by the use of allopurinol had an increase in eGFR of 3.3 ± 1.2 mL/min/1.73 m$^2$ per year, whereas eGFR decreased 1.3 ± 0.6 mL/min/1.73 m$^2$ in the control group during a 12-month follow-up ($p < 0.04$) (*Sezer et al., 2014*). In another double-blind, randomized, placebo-controlled single center trial with CKD stage 3–4 patients, 93 patients with hyperuricemia were randomized to febuxostat or placebo group. The eGFR of febuxostat group increased nonsignificantly from 31.5 ± 13.6 (SD) to 33.7 ± 16.6 mL/min/1.73 m$^2$ at 6 months. Placebo group showed a significant decrease of eGFR from 32.6 ± 11.6–28.2 ± 11.5 mL/min/1.73 m$^2$ ($p < 0.003$) (*Sircar et al., 2015*). Thus, lowering serum uric acid level could slow the decline in eGFR in CKD stages and 4 patients.

In contrast with studies that were done in patients who already have chronic kidney disease, our results show that in health check-up individuals, the increase in SUA level is associated with a greater decline in GFR in a 5-year period. This is in consistent with previous studies that show lowering uric acid level also could preserve GFR in individuals with normal kidney function. Serum uric acid level >480 umol/L (8 mg/dL) or <300 umol/L (5 mg/dL) was found to be linked with a subsequent increased risk for kidney failure within 2 years (2.9-fold in men and 10.0-fold in women) in a study of 6,400 individuals with normal kidney function (*Iseki et al., 2001*). Another three months of prospective study showed that lowering serum uric acid level in patients with normal kidney function could preserve eGFR (*Kanbay et al., 2007*). Taken together, a beneficial effect of controlling SUA level in general asymptomatic population is worthy investigated further.

In our study, we focused on the relationship between SUA level changes and eGFR changes, while the predictive value of hyperuricemia on the incidence of CKD or Cardiovascular Disease (CVD) was not explored. SUA as a risk factor for the incidence of CVD was clearly shown by two recent meta analysis (*Braga et al., 2016*; *Li et al., 2016*). In one residential cohort, SUA and eGFR was separated assessed as risk factors for CVD. SUA level was shown to be a predictor of incidence of CVD while eGFR was not a predictor (*Puddu et al., 2014*). Since the incidence of CVD is higher in CKD especially ESRD patients, the impact of SUA change and eGFR change on the incidence of CVD need further studied.

The mechanisms that uric acid leads to CKD progression are not fully understood. Uric acid crystals are able to adhere to the renal epithelial cells, which could induce an acute inflammatory response (*Umekawa, Chegini & Khan, 2003*). Besides the increased risk of kidney stone formation, uric acid adherence has been shown to reduce the GFR decades ago (*Spencer, Yarger & Robinson, 1976*). Previous studies demonstrated the mechanisms by which that uric acid damages renal function include activation of cytoplasmic phospholipase A2, inflammatory transcription factor nuclear factor-kb (NF-κB) (*Han et al., 2007*) as well as upregulation of the systemic cytokine production, such as tumor necrosis factor α (*Netea et al., 1997*), and the local expression of chemokines, such as monocyte chemotactic protein 1 (*Kang et al., 2002*). Increasing uric acid levels also could induce oxidative stress and endothelial dysfunction.

There are limitations in our study. The major limitation of the present study is that our study subjects were not randomly sampled. Thus, some selection bias cannot be excluded in the cohort. This might explain that some continuous variables in our study are not normally distributed. Second, there are no information on medications used, such as uricosuric drugs, xanthine oxidase, and anti-hypertension drugs. So there are maybe some other confounding factors that could not be detected by our analysis. Third, some subjects showed increased or decreased eGFR that is greater than 50 mL/min/1.73 m$^2$ from 2008–2013 (Fig. 1), which might reflect undetected renal disease. Finally, this study didn't account for all the factors that might have impact on ΔeGFR such as treatment.

In conclusion, in this retrospective study, we observed that larger decline of eGFR over years is associated with the increased SUA level in general population. Together with more causal mechanisms to be discovered, our findings help to underline the importance of SUA level management in the context of kidney function preservation.

## ACKNOWLEDGEMENTS

We thank Shumei Wang and Mingyu Luo for the assistance in statistical analysis.

### Funding

This work was supported by the National Natural Science Foundation of China (81571911, 81171818 to YW) and Grant BS2015YY018 from Shandong Provincial Natural Science Foundation to YX. The funders had no role in study design, data collection and analysis, decision to publish, or preparation of the manuscript.

### Grant Disclosures

The following grant information was disclosed by the authors:
National Natural Science Foundation of China: 81571911 and 81171818.
Shandong Provincial Natural Science Foundation: BS2015YY018.

### Competing Interests

The authors declare that they have no competing interests.

### Author Contributions

- Ying Xu conceived and designed the experiments, analyzed the data, wrote the paper, prepared figures and/or tables, reviewed drafts of the paper.
- Xiang Liu analyzed the data, contributed reagents/materials/analysis tools, reviewed drafts of the paper.
- Xiaohe Sun contributed reagents/materials/analysis tools, reviewed drafts of the paper.
- Yibing Wang conceived and designed the experiments, reviewed drafts of the paper.

### Human Ethics

The following information was supplied relating to ethical approvals (i.e., approving body and any reference numbers):

This work was approved by the Ethics Committee of Shandong Provincial Hospital Affiliated Shandong Provincial Hospital (Approval Reference No. 2015-040).

### Data Deposition

The raw data has been supplied as a Supplemental Dataset File.

### Supplemental Information

Supplemental information for this article can be found online at http://dx.doi.org/10.7717/peerj.1859#supplemental-information.

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
