# Peer review of "The impact of serum uric acid on the natural history of glomerular filtration rate: a retrospective study in the general population"

_PeerJ, doi:10.7717/peerj.1859_

## Round 0.1 · original submission · Major Revisions

· Academic Editor

Major Revisions

As you can see, we obtained what may look like contradictory reports, since one reviewer suggested plain rejection whereas another thought that only minor corrections are needed.

Having examined carefully your paper in the light of the reviewers' comments, I have come to the conclusion that both reports point to important deficiencies that could, however, be corrected by paying very close attention to all of the remarks raised.

If you think you can undertake this work, we'll gladly consider examining a revised version. You should, however, provide a detailed reply to each of the points raised.

·

Basic reporting

The idea is of interest.

Experimental design

There are problems with the eGFR formula selected: at least comment.

Validity of the findings

There should be smoothing on conclusions and more discussion on the relavance as external predictor of SUA.

Comments for the author

Ying Xu et al. from the Department of Nephrology, Shandong Provincial Hospital Affiliated to Shandong University, Jinan, China, considering that serum uric acid (SUA) level has been proposed to have important connections with chronic kidney disease (CKD), while the impact of SUA level on the natural history of glomerular filtration rate (GFR) decline remains unknown investigated, among 2789 subjects who visited the Health Checkup Clinic both at 2008 and 2013, the association of the SUA level with the GFR decline. They observed a significant inverse correlation between change in SUA from 2008 to 2013 (ΔSUA) and change in eGFR (ΔeGFR ) during the same period. Multivariate regression analysis of ΔeGFR indicated that the increase in SUA over time were negative predictor of the change in eGFR. Based on these results they concluded that “the decline of eGFR over years is larger in subjects with an increased SUA level, suggesting a beneficial role of asymptomatic hyperuricemia management in the context of kidney function preservation”.

I am not convinced that from an association study one may correctly derive a cause and effect relation as the one reported in their conclusions which should be modified and smoothed.

Although the Authors concentrated on the relation between SUA changes and those of eGFR and they did not study the effects of both agents on the predictive capacity these factors may have on the incidence of CKD or CVD events, I believe they should have commented the results of investigations such as “Serum uric acid and eGFR_CKDEPI differently predict long-term cardiovascular events and all causes of deaths in a residential cohort. Int J Cardiol 2014; 171: 361-367.“, inasmuch as in the latter study a really residential cohort was explored and they abundantly quoted the study by Chuang et al. „Hyperuricemia and increased risk of ischemic heart disease in a large Chinese cohort. Int J Cardiol 2012; 154: 316-321.” Whereby the predictive role of hyperuricemia for CHD was explored.

Estimated glomerular filtration rate (eGFR) was calculated using Modification of Diet in Renal Disease (MDRD) formula for Chinese. Whereas this may be appropriate in consideration of the population used, it should be at least important to discuss whether comparisons were done, in Chinese populations, with the CKDEPI, more appropriate for these calculations among non-hospitalized individuals.

Correct error 8.36, line 142: ΔeGFR (r=8.36, r=0.15, r=0.10, r=0.09, r=-0.30, respectively; p< 0.01).

The assumptions at lines 170-1 „Our results suggest that if the SUA were well controlled over time, the eGFR would decline slower.” Is hypothetical and should be eliminated.

An aspect that should be discussed, apart the reciprocal role of deltaSUA and deltaeGFR on predicting CKD (or just renal function deterioration) is which one between them is the most important predictor of renal function in terms of “external” predictivity and, more precisely, how this could impact on the predictive role that kidney function has for CVD outcome in the broder sense from CHD to stroke and peripheral arterial disease.

Reviewer 2 ·

Basic reporting

Novelty is not enough and poor organization

Experimental design

not high quality

Validity of the findings

not good

Comments for the author

Dr Xu and colleagues report a study to evaluate the impact of serum uric acid on the natural history of glomerular filtration rate among a general population in a retrospective cohort. This is a relatively large sample size study in one center for more than five years and time varying data was obtained in eGFR and serum uric acid (SUA). It concludes that the change in SUA during a 5-yearperiod is inversely associated with the decline of eGFR during the same period. But there are some major concerns on this manuscript should be addressed by authors.

1. There were at least 10 studies analysed data from population-based participants who underwent health examinations, and reported the relationship between increasing serum uric acid level and the development of eGFR< 60ml/min (new-onset CKD) during follow-up of non-CKD participant. Among them, nine studies were conducted in Asian population(Li et al. BMC Nephrology 2014, 15:122). Meta-analysis showed that with long-term follow-up of non-CKD individuals, elevated serum uric acid levels showed an increased risk for the development of chronic renal dysfunction1. As such, this manuscript also did not include enough important references in the introduction, a more comprehensive review of the literature data should be done.

2. Which formula was used for the calculation of eGFR in this study, please give the references, furthermore, as no age inclusion criteria was mentioned, was any participants younger than 18 years enrolled? If yes, which formula was used for this part of participants?

3.there are some concerns on statistics and data report.

Considering the general population studied, adding this restriction of population in the title of this manuscript would be more appropriate.

In Line 115 to 117 and Table1, paired t tests or McNemar’s tests should be used in comparison between data in year of 2008 and that in 2013.

Large standard deviations of some variables such as SUA, FPG, TG (in Table 1) are observed, and normality test is needed for such kind of data before choosing methods for description and comparison between groups. In Line 159 to 162, data also shows that the deviations of ΔeGFR are too large to be described as means with SD (17.95 vs. 18.81). And in Line 114 and 115, “Descriptive statistics are presented as means and frequencies with standard deviations.” should be that “Descriptive statistics are presented as means with standard deviation for normally distributed data, median with interquartile range with non-normally distributed data, and frequencies with percentage for categorical data”.

In Line 163 to 165 and in Table 3, the authors give the results of multiple linear regression model to assess the impact of ΔSUA on ΔeGFR, however, they are poorly reported and interpreted.

First, R squared, adjusted R squared should be given to show the goodness of fit of this model. And in Line 119 to 120, it is not enough to describe “among the candidate models, we selected the best-fit model for each dependent variable”, but also to explain what “the best-fit” means and which statistics be used.
Second, simple linear regressions should be modeled before doing multiple linear regressions and the author should explain why those variables be selected in the final model. Dose multicollinearity exists in those variables and how the authors avoid it?
Finally, standard coefficients, and p-values with exact number should be given in Table 3.

4. In Line 152 to 153, the authors describes that “We divided the subjects to two groups: people with an increased SUA and people with a decreased SUA.” There maybe people with an unchanged SUA. The author should clearly define the groups.

5. Table2 shows that the standard deviation of the variable of FPG in increased SUA group is much larger than that in the other group (12.08 vs. 1.15), and in Line 142 it reports that r=8.36, which are both unreasonable. The author should carefullycheck them.

6. Conclusion should be more appropriate stated, for example: Abstract part, “Our result indicates that the decline of eGFR over years is larger in subjects with an increased SUA level, suggesting a beneficial role of asymptomatic hyperuricemia management in the context of kidney function preservation.” Notably, the studied subjects with hyperuricemia were not “asymptomatic”, as mentioned in Methods part (line 82:This study excluded outpatient or clinical patients, patients with acute kidney injury, amputation, heart failure, severe liver disease, infection disease, malignant disease and pregnancy were excluded) .

---

## Round 0.2 · accepted · Accept

· Academic Editor

Accept

Your paper was reexamined by one the reviewer who assessed your first submission. His opinion was that you satisfactorily answered most of the questions and met the remarks of the reviewers. The other reviewer declined to reexamine your work.

·

Basic reporting

I have appreciated the attention paid by these Authors to the comments raised by both Referees and the changes introduced in the revised MS.

Experimental design

The changes introduced and the new analyses performed have improved this MS.

Validity of the findings

These are valid data and the Authors have correctly pointed out in Limitations some of the shortcomings that may apply.

Comments for the author

I congratulate these Authors for their interesting study opening up on the role of intervening on uric acid to keep renal function in good health over time.